# Towards Dynamic Multi-Modal Intent Sensing Using Probabilistic Sensor Networks

**DOI:** 10.3390/s22072603

**Published:** 2022-03-29

**Authors:** Joseph Russell, Jeroen H. M. Bergmann, Vikranth H. Nagaraja

**Affiliations:** Natural Interaction Lab, Department of Engineering Science, Institute of Biomedical Engineering, University of Oxford, Parks Road, Oxford OX1 3PJ, UK; jeroen.bergmann@eng.ox.ac.uk (J.H.M.B.); vikranth.harthikotenagaraja@eng.ox.ac.uk (V.H.N.)

**Keywords:** bionic prostheses, intent sensing, Bayesian sensor fusion, body sensing, wearable technology, sensor networks

## Abstract

Intent sensing—the ability to sense what a user wants to happen—has many potential technological applications. Assistive medical devices, such as prosthetic limbs, could benefit from intent-based control systems, allowing for faster and more intuitive control. The accuracy of intent sensing could be improved by using multiple sensors sensing multiple environments. As users will typically pass through different sensing environments throughout the day, the system should be dynamic, with sensors dropping in and out as required. An intent-sensing algorithm that allows for this cannot rely on training from only a particular combination of sensors. It should allow any (dynamic) combination of sensors to be used. Therefore, the objective of this study is to develop and test a dynamic intent-sensing system under changing conditions. A method has been proposed that treats each sensor individually and combines them using Bayesian sensor fusion. This approach was tested on laboratory data obtained from subjects wearing Inertial Measurement Units and surface electromyography electrodes. The proposed algorithm was then used to classify functional reach activities and compare the performance to an established classifier (k-nearest-neighbours) in cases of simulated sensor dropouts. Results showed that the Bayesian sensor fusion algorithm was less affected as more sensors dropped out, supporting this intent-sensing approach as viable in dynamic real-world scenarios.

## 1. Introduction

### 1.1. Intent Sensing

The idea of measuring a person’s “intent”—broadly, what they want to volitionally make happen in the physical world at a given moment—has great potential for the development of technology involving human–system interaction. Most currently available devices, from personal computers or mobile phones to home virtual assistants, require direct cues from the user for control. This might be through a touch interface, a voice command, or even a gesture, but all require a conscious input from the user to operate the device [1].

Conversely, a device controlled through intent sensing would not need a direct input from the user. Instead, with networked (non-invasive) sensors, the user’s needs could be passively measured or even anticipated, and the device of interest could be activated automatically.

An example of this could be by using a person’s body language, such as the fact that they are sitting on a sofa or the inclination of their head towards the television, combined with contextual information such as the time of day and the user’s usual routine, to infer that the person would like to watch their favourite programme. An intent-sensing smart television could then turn on and even select the programme without the need to operate a remote control.

A less trivial benefit of this kind of technology, however, would be for use in medical devices, where the accurate understanding of the user’s intent could help improve their quality of life. The potential applications are wide-ranging, including assistive robotics for stroke patients [2] and detecting physical activity for personalised drug delivery in those with diabetes [3]. One application of particular interest is prosthetics.

As of 2017, there are estimated to be 57.7 million people worldwide living with limb amputation due to traumatic causes [4], many of whom use or are in need of a prosthetic device. One of the principle causes of device abandonment is reported to be difficulty of use [5], and as such, an intent-sensing system that could improve the intuitiveness and user experience of a prosthetic device has the potential to be extremely valuable.

According to Lee et al. [6], intent sensing is composed of three aspects: (i) the recognition and identification of activity transitions, (ii) the inference of task goals, and (iii) the prediction of future activities. See Figure 1 for an illustration of this.

Current intent-sensing literature contains a well-established body of work related to activity transition recognition, with less research on task goals and predictions [1]. This paper will predominantly focus on transition recognition, followed by an inference of the task goal. While attempts will be made to perform classification as early on as possible in an activity, outright prediction before an activity begins is not within the scope of this study.

### 1.2. Sensor Networks for Intent

The accuracy of input detection is particularly essential in assistive medical devices such as prosthetics, where errors could lead to both frustration and injury. Provided effective sensor fusion algorithms are used, accuracy can be improved by combining information from multiple sensors. To produce the highest possible accuracy, the intent-sensing system should take advantage of all sensors available at any time.

A range of embedded sensors might be available in a typical prosthetic device. These can often include surface electromyography (sEMG) sensors integrated into the socket, measuring electrical activity in the residuum of the user’s superficial muscles [7]. Kinematic sensors are also often available, providing measurements of the device’s orientation and acceleration through Inertial Measurement Units (IMUs) and combining together information from accelerometers, gyroscopes and magnetometers [8].

Multiple sensors could be added to a prosthetic to increase accuracy. However, there are potential disadvantages to adding more and more sensors, such as increases in cost, weight, power consumption and difficulty donning/doffing the device. To further improve the accuracy of the system without introducing these issues, information from external sources could also be included.

Smart phones and smart watches, for instance, include a whole range of sensors, such as audio and GPS sensors and are commonly used throughout the day. When these devices are in use, they provide additional sensory information largely independent from the sensors that are embedded in the medical device [9].

Smart home technology is also becoming increasingly prevalent [10], including smart doorbells, speakers, home assistants and more. In-home monitoring of patients using (both wearable and contactless) sensors for medical purposes is also an increasingly common practice. Smart environments are also being developed in workplaces, on public transport and within private vehicles, each providing a range of sensory information that could be used to provide predictions of intent.

These are examples of how device users pass through varying sensor-rich environments, with sensors becoming available or unavailable throughout the day. To take advantage of these uncertain information sources, an intent-sensing system should be dynamic, following a “drop-in/drop-out” structure, incorporating information from sensors only when they are available and weighting their contribution according to their accuracy, which would have to be pre-learned on a sensor-by-sensor basis rather than as a complete, fixed network. This approach would allow for any set of sensors to be combined for intent sensing.

This would not only enhance the performance of the system by utilising all available resources whenever possible, but would also improve system robustness in case of the failure of some of the sensors. A system that is still able to run (albeit with reduced accuracy) with only a subset of its sensors will have a huge advantage over systems that require all their sensors to work in unison to function.

This study therefore proposes to investigate the use of a new, modular approach to networked intent sensing that is not trained on any particular combination of sensors, but instead is able to freely add in or remove sensors to produce robust intent predictions.

### 1.3. Probabilistic Sensor Networks

One approach to creating the proposed robust, drop-in/drop-out system is to model the process as a Probabilistic Sensor Network (PSN) [11]. This method breaks the process of detection down into four stages, each with their own independent probability of the signal correctly passing through them: Environment, Sensing, Conditioning and Processing (See Figure 2). Assuming independence between the stages, the total probability of correctly detecting an event is the product of these four probabilities.

Since the probability associated with each stage is naturally less than or equal to 1, the total probability of correctly detecting an event (in this case, a particular user intent) will never be greater than the highest probability of the four stages. Therefore, each stage acts as a potential “bottleneck” for the system. For instance, if the probability of an intent being detectable from the electrical activity in the muscle is only 0.8, then the Environment probability is 0.8, and so it does not matter how accurate the Sensing, Conditioning or Processing stages become—the total probability will never exceed 0.8.

Each stage can contain multiple sensing nodes. Adding a second sEMG sensor to another site on the muscle and combining the two sensors together with a sensor fusion algorithm increases the probability of the Sensing stage. However, both sensors are operating in the same Environment, and so the total probability will still be limited to 0.8. Adding sEMG sensors to a second Environment, such as a different muscle, helps circumvent this “bottleneck” and allows for the total probability of the system to increase.

In order to keep the network dynamic without being limited to any particular configuration of sensors, each sensor node will be considered on its own and only combined together at the final Processing stage.

In the proposed intent-sensing system, sensors that are added may have very high accuracies for identifying some intents and low accuracies for others—such as sEMG sensors placed on muscles in the lower left leg when used to detect walking versus using the same sensors to detect a reaching motion of the arm. A simple majority voting system would not account for this, so it would be unsuitable.

Instead, a more versatile Bayesian approach can be taken. Where P(E) is the probability of the event occurring and P(V) is the probability of getting a particular set of sensor values, Bayes’ rule [12] gives the probability that an event has happened given a set of sensor values, P(E|V), as:(1)$$P\left(E|V\right)=\frac{P(V|E)\cdot P\left(E\right)}{P\left(V\right)}$$

The dataset probability can be obtained using the total probability rule [13], and so the confidence can be written as follows, where $P\left(E^{\prime }\right)$ represents the probability of the event not occurring and $P\left(V|E^{\prime }\right)$ represents the probability of getting a set of sensor values given that the event has not occurred:(2)$$P\left(E|V\right)=\frac{P(V|E)\cdot P\left(E\right)}{P\left(V|E\right)\cdot P\left(E\right)+P\left(V|E^{\prime }\right)\cdot P\left(E^{\prime }\right)}$$

P(V|E)  is obtained from the sensor sensitivities. It is the probability of getting a specific set of sensor values given that the event did occur. If they are independent, this is the product of their probabilities. P(E) is a prior for the probability of the event occurring and could be obtained from contextual information, such as user routine and time of day.

Calculating a confidence value in this way for each possible intent and choosing the intent option with the highest confidence is a more effective method of combining sensor outputs. Effectively, it is giving an optimal “weighting” to each sensor’s contribution according to its individual accuracy. Combining sensor information according to this method means that, provided the accuracy of each sensor is precisely known, adding sensors can only monotonically improve the overall accuracy of the system, even if only by a very small amount as in the case of sensors that are close to random in their predictions.

To apply such a method, accuracy estimations for each sensor will need to be obtained for each possible intent option, i.e., each sensor will require a known confusion matrix. This should list the probability of each intent being true given the sensor’s prediction of a particular intent. The entries for this confusion matrix can be populated through calibration. This can be completed individually for each user, for all users or for some combination of the two, starting with a general estimation and adjusting it to become more personalised over time.

In the case of sensors dropping in and out over time, all that is required in this Bayesian method is for their confusion matrices to be added to or removed from the equation. No retraining of the other sensors is needed to compensate for the change, and therefore this method is interesting as a potential solution to the need for a dynamic network.

Other, more advanced methods of combining sensor measurements together do exist. Many of these employ machine learning techniques, such as decision trees [14], random forest classifiers [15] and support vector machines [16]. These techniques do not treat each sensor as an individual “black box” and instead consider, for example, the relationships between sensors. Exploiting this extra dimensionality has the potential to provide additional information, suggesting it is a more suitable technique. However, it also means that they must be trained on specific combinations of sensors and that adding and removing dimensions from the trained classifiers “on the fly,” as sensors drop in and drop out of the network, is not within the capability of established machine learning techniques (if not trained for it).

It is therefore proposed that while combined machine learning techniques are theoretically able to perform better than the Bayesian fusion technique, they will rapidly drop in performance when sensors are removed, and repeated experiments simulating sensors dropping out should show this effect.

### 1.4. Objective

The objective of this study is to develop a Modular Method (MM) suitable for the dynamic environments described previously and to compare it to a Non-Modular Method (NMM), utilising a combined machine learning (cML) algorithm representative of current commonly used methods. The experiments performed should measure classification accuracy early in the activity cycle, well before any activity example is completed, in order to demonstrate the goal inference aspect of intent sensing. The change in accuracy with increasing time allowed between the activity’s inception and classification should be determined.

Then, the two algorithms should be compared in their accuracy versus a varying number of sensors when they were allowed to train on the exact combination of sensors they are being tested on. The hypothesis is that the new MM will not perform as well as the NMM.

Subsequently, a scenario of random sensor dropout will be introduced, where the sensor combinations required are not trained on in advance. The hypothesis is that the MM will perform better overall than the NMM, with the difference in accuracy between the two techniques increasing as more sensors drop out.

## 2. Materials and Methods

### 2.1. Data Collection

Data used for this study were originally gathered under laboratory conditions as part of a prior study [17]. All data were anonymised, and prior informed consent was obtained from each individual. The study was approved by the institutional ethics committee (Reference Numbers: 16/SC/0051 and 14/LO/1975).

Five adult non-disabled participants wore full-body MVN-Awinda [18] Inertial Measurement Units (IMUs) (Xsens Technologies B.V., Enschede, The Netherlands) during task execution (sampling frequency: 60 Hz). Passive retro-reflective markers were placed on their body, which were tracked with a 16-camera Vicon (Vicon, Oxford, UK) motion tracking system. This Optical Motion Capture (OMC) data (sampling frequency: 100 Hz) were used for the verification of activity labels.

A wireless 10-channel Zerowire (Aurion Srl, Milan, Italy) electromyography (EMG) system was used to collect surface EMG (sEMG) data at a sampling frequency of 1000 Hz for the five selected superficial muscle groups: Pectoralis major (Clavicle), Biceps brachii, Triceps (Long head), Deltoid (Medial) and Brachioradialis.

Subjects were asked to perform three trials each of eleven different reach/grasp activities (Reach to grasp: (i) forward, (ii) left, (iii) right and (iv) up; Reach: (v) forward, (vi) left, (vii) right and (viii) up; (ix) hand to mouth, (x) hand to the top of head and (xi) hand to contralateral shoulder), which were grouped into three categories: Reach tasks, Reach-to-Grasp tasks and Gross Motor Skill tasks.

These actions were performed using a specially built test rig, shown in Figure 3. A total of 165 datasets were collected. Each participant also performed a Maximal Voluntary Isometric Contraction (MVIC) test with each measured muscle group to allow for normalisation of the corresponding experimental sEMG data.

To ensure the experiment was representative of intent sensing rather than pure activity classification, only the first 1000 ms of each dataset were used.

### 2.2. Processing

The affiliated Xsens MVN Analyze software [19] was used to process the IMC data and export it as MVN open XML (.MVNX) files to be used as inputs for the algorithms.

The OMC data corresponding to marker trajectories was processed and exported in C3D or Coordinate 3D (C3D.ORG; https://www.c3d.org/index.html; accessed on 26 February 2022) format using the Vicon Nexus 2.5 software [20]. Further processing took place in MATLAB R2020b (Mathworks, Natick, MA, USA). The sEMG data was synchronised with the IMU data from the XSens IMC system using the recommended protocol [21].

The sEMG data was filtered using a 10–500 Hz band-pass fourth order Butterworth filter and normalised according to the maximum signal measured in the MVIC tests [22].

### 2.3. Feature Extraction

A breakdown of the features extracted from the IMU and sEMG sensor signals is shown in Table 1.

The three IMU channels included were orientation, accelerometer and magnetometer signals. The raw values for these were used directly in the training step. These three distinct data channels allowed each IMU sensor to be more closely approximated as probabilistically independent by the Bayesian model. The orientation measurement contained some co-dependency on the accelerometer and magnetometer readings.

For the sEMG signals, a more complex feature extraction method was required. The process detailed in [23] was followed to ensure standard methods were applied. The data for each participant were divided into 200 ms segments and shifted by 50 ms increments (such that consecutive segments overlapped by 150 ms). Within these segments, the following features were extracted: Integrated EMG, Mean Absolute Value, Mean Absolute Value Slope, Variance of EMG, Root Mean Square, Waveform Length, Autoregressive Coefficients (to the fourth order), Frequency Median and Frequency Mean.

This full set of features was carried forward for analysis. An investigation of feature reduction, not used in the final algorithm, may be found in Appendix A.

### 2.4. Data Separation (MM/Bayesian Fusion Only)

To allow as large a training set as possible, leave-one-out cross-validation was used. As such, for each repetition of the analysis, one sample was held back for testing (the Testing Set), leaving 164 samples for training.

In order to learn the probabilities associated with each sensor and thereby populate each sensor’s confusion matrix (as is required for the Bayesian sensor fusion method) without introducing any element of bias, the remaining samples were divided again. Half of the 164 samples were pseudo-randomly selected and used to train an MM classifier for each sensor (the Classifier Training Set), and the remaining half were used to test the MM classifiers and measure their accuracy for each activity (the Probability Learning Set). To complete this, the number of successful classifications was divided by the number of samples for each activity to estimate the probability, which was recorded in the confusion matrix. The sums of the diagonal entries of the confusion matrices were then used to approximate each sensor’s overall accuracy.

The choice of which samples were placed in the Classifier Training Set and which in the Probability Learning Set could have an impact on the performance of the classifier. To produce results representative of all subjects and activities, a selection algorithm was used to pseudo-randomly place an approximately equal number of data samples from each participant and activity type in the Classifier Training and Probability Learning Sets. Where multiple examples were available (each participant provided three samples of each activity), the set they were placed in was randomly selected.

An optimisation step also took place here, repeating the previous steps five times, with a different pseudo-randomly selected and split between the sets each time. The classifier chosen to take forward was whichever resulted in the highest mean accuracy across sensors, as measured in the Probability Learning Set.

For the comparison, a Combined KNN technique was used in the NMM; this step was not required, as all data in the training set from all sensors were used to train a single classifier, with no estimation of the classifier’s accuracy.

### 2.5. Learning Classifiers

For both the MM (Bayesian Fusion algorithm) and the NMM (Combined KNN algorithm), KNN classifiers were used. The distinct difference between the two was that, for the MM, one KNN classifier was trained for each sensor and then combined, whereas the NMM trained a single classifier using all the sensors as inputs (see Figure 4 for a graphical representation).

The features described in Section 2.3 were used as inputs to train the KNN classifier [24], with hyperparameter optimisation selecting an N value (no. neighbours) of 1 and a Gaussian distance metric with an exponent of 0.5. The same number of features was used for both algorithms. For the Combined KNN classifier, all features were used for one algorithm, whereas for the Bayesian Fusion algorithm, the features were evenly distributed across the individual sensors.

### 2.6. Sensor Fusion (Bayesian Fusion Only)

In the Bayesian Fusion algorithm, each sensor produced its own independent classification of the activity. Equation (2) was used with the confusion matrices populated in the Probability Learning Set in order to calculate a probability of each activity being the true activity given the sensor values. Whichever activity had the highest probability was selected as the output of the combined system.

For the comparison with the Combined KNN algorithm, all the sensor inputs were fed into the KNN algorithm previously used exclusively within each EMG sensor. This is a well-established supervised learning technique [25,26] used in many EMG-driven intent-sensing studies and should be representative of the general performance of machine learning techniques.

### 2.7. Testing

The performance of the two algorithms was compared by testing their classification of the data sample in the testing set. The total number of correct classifications across the 165-fold leave-one-out cross-validation method was divided by the number of trials (165) to result in an accuracy measure for each algorithm.

### 2.8. Time Variation

The first goal to be investigated was the effect on the accuracy of intent classification when varying amounts of time were allowed to pass after the activity’s inception before intent classification was performed. To measure this, the experiment was repeated with all 24 sensors active, making the prediction using only the first X milliseconds of each sample, with X increasing from 200 in 50 ms increments up to the full 1000 ms allowed. The resulting accuracies were plotted against the time allowed to show the trend. The trend for both methods was then quantified using a Spearman’s rank correlation coefficient, where $R\left(P_{i}\right)$ and $R\left(T_{i}\right)$ are the ranks of each (*i*-th) sample in accuracy and time, respectively, and n is the number of samples:(3)$$r_{s}=1-\frac{6\sum^{\;}{\left(R\left(P_{i}\right)-R\left(T_{i}\right)\right)}^{2}}{n\left(n^{2}-1\right)}$$

This gives a result between −1 and 1, where 1 is a perfectly monotonically increasing pattern, −1 is a perfectly monotonically decreasing pattern and 0 indicates no monotonic relationship. This is an appropriate measure, as it will indicate to what extent the hypothesis is true, namely that accuracy will increase when more time is allowed to pass [27].

### 2.9. Variant No. of Sensors 

The second area to be investigated was the effect on intent classification accuracy when the number of sensors was varied, where in each instance, the algorithms were trained only on the sensors that were active. It should be noted that this is not testing robustness to sensor dropout (this is described in Section 2.10) but instead showing the effect of increasing the number of sensors.

The number of sensors used as inputs, R, was varied from 1 to 24. For each number, 50 randomly selected combinations of R sensors were tested and the mean accuracy was recorded, along with the 95% confidence intervals. These were plotted and compared graphically.

Referring back to the PSN model discussed in Section 1.3, as both algorithms use the same sensors in this experiment, in the same sensing environments and with the same signal conditioning, this suggests the difference in probability comes entirely from the Stage D, Signal Processing step.

If the probability associated with the MM (Bayesian Fusion) method is P(D1) and the probability associated with the NMM (Combined KNN) method is P(D2), and the probabilities of the Environment, Sensing and Conditioning stages are P(A), P(B) and P(C) for both algorithms, respectively, the ratio of the Processing probabilities for the two algorithms is given by:(4)$$\frac{P\left(BF\right)}{P\left(CKNN\right)}=\frac{P\left(A\right)\cdot P\left(B\right)\cdot P\left(C\right)\cdot P\left(D1\right)}{P\left(A\right)\cdot P\left(B\right)\cdot P\left(C\right)\cdot P\left(D2\right)}=\frac{P\left(D1\right)}{P\left(D2\right)}$$

This ratio therefore quantifies the relative benefit of using the MM over the NMM for intent detection, and so this value will be estimated using the data points plotted and then used to compare the algorithms. This ratio is not expected to be consistent for all data points, and so the range of values will be given.

### 2.10. Simulated Dropout

The third (and most important) element to be investigated was the effect on the intent classification accuracy of the algorithms only trained with all sensors active when a number of sensors begin “dropping out”. To test this, the experiment was repeated again, this time by randomly selecting N sensors to be set to a constant 0. For the Bayesian algorithm, these sensors’ predictions were not used. On the other hand, the K-nearest-neighbours algorithm trained on all the sensors cannot have an input removed, so they continued to use the 0 value.

The number of dropped sensors, N, was increased from 0 to 23, with the predictions made only 1000 ms into each activity—well before their completion, making this analysis a “goal inference” task. The accuracies measured were averaged over 50 random combinations of N sensors dropping out. The mean accuracies over twenty repetitions of the MM (Bayesian Fusion) and NMM (Combined KNN) were plotted against N, along with the upper and lower bounds of the 95% confidence interval of each.

As in Section 2.9, the ratio of the accuracy of the MM to the NMM will be used to quantify the relative benefit. This will vary as N increases, and so the range will be given.

## 3. Results

The accuracies of the two classification methods versus an increasing amount of time allowed after activity inception with all sensors used are shown in Figure 5. Both showed trends of increasing accuracy over time, with a Spearman’s rank correlation coefficient of 0.9 for the MM (Bayesian Fusion) and 0.6 for the NMM (Combined KNN).

The accuracies of the MM and the NMM trained and tested on a varying number of sensors from 1 to 24 with no drop out are shown in Figure 6. The NMM had consistently higher accuracy than the MM in this case, with no overlap between the 95% confidence intervals until the number of sensors reached 24, at which point the upper confidence interval of the MM exceeded the lower confidence interval of the NMM. The accuracy ratio of the MM to the NMM method ranged between 0.8 and 0.9, with a mean of 0.85.

The accuracies of the MM and the NMM all trained on the complete set of sensors when the number of sensors dropping out varies from 0 to 23 are shown in Figure 7. Initially, with no sensors dropping out, the NMM resulted in a higher mean accuracy than the MM. As more sensors dropped out, this difference decreased, until with 10 sensors dropping out, the MM mean accuracy exceeded that of the NMM. From this point on, the accuracy advantage of the MM over the NMM continued to increase, resulting in an accuracy ratio of MM to NMM ranging from 0.9 at 0 dropout to 1.4. Initially, there was a large amount of crossover between the 95% confidence intervals of the two methods, but after 17 sensors had dropped out, the lower bound of the MM became greater than the upper bound of the NMM, and remained so for all greater numbers of dropped sensors.

## 4. Discussion

### 4.1. Time-Dependent Classification

The results obtained from the experiment demonstrated a link between the time allowed to collect data and the accuracy of the classification, with the accuracy increasing approximately monotonically over time (see Figure 5). This link was shown more strongly for the MM than for the NMM, with a higher Spearman’s correlation coefficient for the former than the latter. There is no reason in the algorithms that the time allowed should have a different effect on the MM than the NMM. It is possible that the difference is due to the experiment being performed with all 24 sensors active—in this case, the NMM is expected to perform better than the MM, and is therefore closer to the maximum accuracy achievable by the sensors. As a result, there is less capacity for an increase as more time is allowed. Regardless of the difference, the experiment indicates that intent classification is more difficult earlier on in the activity cycle.

This is an expected result, as the activities all start in approximately the same position, with the differences between them increasing as they progress. Similar findings were shown in [28], though in a somewhat different context, and when viewed in combination with this study, it is suggested that the monotonic increase in classification accuracy begins even before activity inception (where it is purely predictive) and continues throughout the activity cycle.

### 4.2. Variant No. of Sensors

A clear relationship is also seen between the total intent classification accuracy and the number of sensors used as inputs in the algorithms (see Figure 6). It should again be noted that, in this test, the algorithms were trained on each possible combination of sensors, which is not possible in real-world applications where the sensors that may or may not be available are not known in advance.

For both algorithms, the classification accuracy increased with the number of sensors. However, as there is no “sensor drop out” in this scenario, the NMM (Combined KNN algorithm) showed consistently better performance than the MM (Bayesian Fusion algorithm).

Again, this is an expected result that aligns with previous studies [29], as the Combined KNN algorithm is able to exploit relationships between the features from different sensors, whereas the Bayesian Fusion algorithm is limited to only exploiting relationships between features within each modular sensor. The Combined KNN algorithm therefore has access to more information than the Bayesian Fusion.

The ratio between the accuracy of the MM and NMM ranged between 0.8 and 0.9, suggesting that while the modular approach does show lower performance than the non-modular approach with no sensor dropout, this difference is relatively small, and much of the accuracy is still retained.

Both algorithms followed a similar pattern of initially rapid increases in accuracy as the number of sensors increases, with a decreasing slope as the number of sensors becomes large, approaching an asymptote. This follows the general pattern expected by a PSN with increasing sensor numbers [30].

### 4.3. Simulated Dropout

The case of simulated dropout is the one most directly pertaining to the proposed real-world application, and is the main focus of this study. While the Bayesian Fusion algorithm was not designed to outperform standard combined methods in situations where the specific combination of available sensors is known and trained on, this pre-training will not be possible in dynamic real-world scenarios.

The situation of sensors dropping out reflects the fact that, in a dynamic sensing environment, sensors which were initially available will no longer become available as the user moves away from them. It also applies to the issue of maintenance, where sensors on a user’s device may fail over time with use, often requiring regular follow-up appointments [31], which may be costly and unfeasible in developing countries [32].

By treating the individual sensors as modular, the Bayesian Fusion algorithm allows any combination of sensors to still function together as a system, rather than relying on all of them. In this scenario, therefore, both algorithms were trained on all available sensors, and then increasing numbers of sensors were randomly set to 0. The hypothesis was that the modular Bayesian Fusion approach would be much more robust to this dropout than the Combined KNN approach, which was dependent on all sensors working together and should therefore drop in accuracy more rapidly.

This was supported by the results, which showed that while the NMM was superior with no dropout (as expected from previous studies [33]), as the number of dropped sensors increased, the MM overtook its accuracy and became increasingly superior. Given that a real-world intent-sensing system might involve hundreds of different sensors dropping in and out throughout the user’s activities of daily life, these results indicate the MM as a more appropriate choice than non-modular alternatives.

Furthermore, an NMM would have to be pre-trained on every possible combination of sensors, which becomes prohibitively complex and computationally expensive with hundreds of sensors in play. Conversely, an MM would only need to train a single classifier for each sensor.

### 4.4. Limitations of the Study

While the algorithms used in this study were designed to still be valid for real-time application, the tests described were performed on data “after the fact.” A practical application would require real-time testing, factoring in elements such as processing speed.

This study involved only five participants, each performing three trials. While this was sufficient to demonstrate the general ideas discussed, small datasets with high dimensionality can lead to bias in performance estimates and inaccuracies in classifiers [34]. As such, a larger dataset would allow the study of more effective classifiers, with more accurate confusion matrices and more precise final accuracy measurements.

This study classified intent between three different classes, but in daily life, intent options are far more diverse.

This study used only IMU and EMG sensors as inputs, in order to demonstrate the combination of different sensing environments for networked intent sensing. Many other sensor types would also be valid.

In terms of sensor numbers, this study was limited to 24 sensors. While this is far more than may be found on a typical prosthetic device, compared to a network composed of wearable, smart-phone and smart home sensors across multiple environments including home, travel, work, etc., it is relatively small. Given the pattern established by this study, it is likely that the difference between the algorithms will only increase with larger numbers of sensors dropping out.

This study only concerned itself with the scenario of sensors dropping out, and did not consider the possibility of sensors dropping in. It would have been impossible to include sensors in the NMM that were not originally trained on, as this would result in more input dimensions than the trained classifier allows. The MM, however, would have had no issue with this, so long as confusion matrix entries for the new sensors were provided.

### 4.5. Suggested Future Work

While the data set used for this work was a good starting point, applying the methods established here to a much larger data set (such as [35]) would also be of interest, allowing the study of more accurate classifiers and more accurate estimations of the confusion matrix entries.

Secondly, the analysis performed here took place offline, with all data recorded in advance. The next stage of the algorithm’s development would be to adapt it to run in real time, perhaps in a scenario similar to [36], which would be much more representative of practical use in a prosthetic device.

More sensors, particularly from multiple sensing environments, would further improve the algorithm and allow more precise classification into finer, more detailed classes. A future study could combine the sensors used in this study (perhaps built into a wearable device) with smart home sensors and built-in sensors on devices such as smart phones and smart watches to further expand the network and provide a better simulation of the proposed real-world application.

The MM also has the benefit of reducing the dimensionality of the classifiers trained compared to the NMM. The NMM, using the same number of training samples, must train a classifier to distinguish between features from every sensor at once. The MM splits this problem up into R individual classifiers (where R is the number of sensors), each with the number of dimensions reduced by a factor of 1/R. It is possible that this could reduce the amount of training data required to train the needed classifiers—a future study could investigate, verify and quantify this potential advantage.

Finally, the maintenance applications of the modular algorithm could be further explored by applying the algorithm to the number and types of sensors found on actual prosthetic devices for current industry-standard classifications, to investigate advantages in continued prosthetic viability as sensors cease to function.

## 5. Conclusions

Firstly, it has been shown that intent classification is easier the later on in the activity cycle it is attempted, and that a high classification accuracy (~96%) can be reached using only the first 1000 ms of data after activity inception for simple tasks.

Adding more sensors has been shown to produce a strong improvement in accuracy regardless of which intent-sensing algorithm was used, well beyond the relatively small number of sensors typically used in prosthetic devices. Networking larger numbers of sensors together is supported by this study as a potential method for improving device input detection accuracy, which could not only lead to better performance in devices but also allow the option of more precise, complex input actions that previously have not been detectable with a high-enough accuracy through existing methods.

The proposed modular approach to sensor fusion supports a dynamic intent-sensing network more effectively than the comparison combined approach, with higher accuracy under conditions of major sensor dropout and the possibility for “drop-ins”, which are not viable for a non-modular system.

This approach is not only useful in a dynamic context where users move from one sensing environment to the other, but also in maintenance scenarios, to allow devices to continue to function with a reduced set of sensors where repair is costly or impossible. It could even be possible to use a combined algorithm where all sensors are available, and then switch to a modular algorithm when sensor dropout is detected, maximising accuracy with a “best of both worlds” approach.

In summary, this is an early exploration into the requirements and viability of a dynamic intent-sensing system, and it is hoped that subsequent research will push this technology further, towards a future where intent sensing approaches ubiquity in medical devices and beyond.

## Figures and Tables

**Figure 1 sensors-22-02603-f001:**
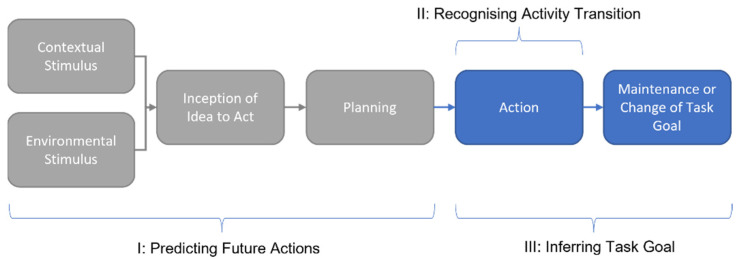
Framework representing the three aspects of intent sensing. Adapted from [1]. Grey regions indicate stages not utilised in this study. Environmental Stimulus indicates change in the user’s surroundings (such as sound or the arrival of another person) that may trigger a response. Contextual Stimulus includes wider factors such as the time of day, typical routine, previous actions etc. The Inception of the Idea to Act represents the user’s conscious decision to take action and predominantly concerns the brain and nervous system. The Planning phase includes any preparation that may take place before the activity begins (such as a visual inspection of the path ahead when about to start walking or the pre-tensing of muscles before attempting a timed grasp). The Action phase covers the real-time execution of the activity, including any changes observable while the activity is being performed. The Maintenance or Change of Task Goal phase looks ahead to the objective of the activity, considering why it is being performed and whether this objective alters over the course of the activity.

**Figure 2 sensors-22-02603-f002:**
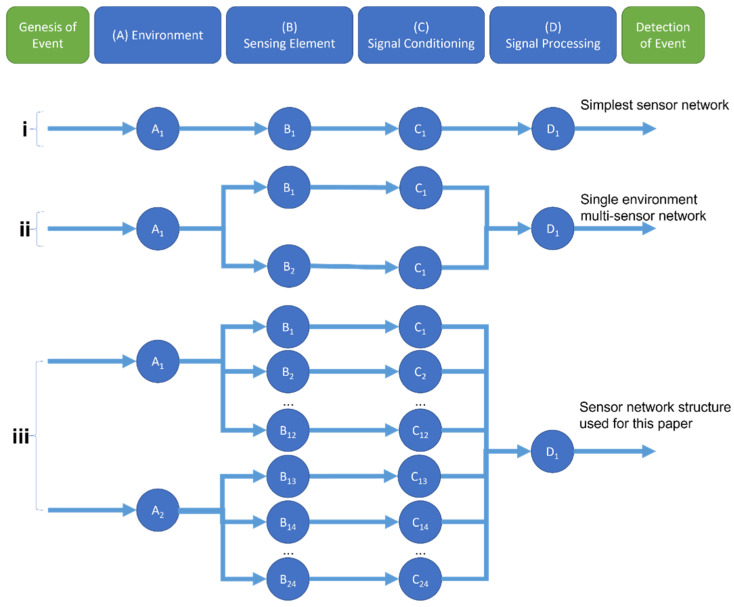
Model of a Probabilistic Sensor Network for event detection (adapted from [11]). The four steps that lead from the genesis of a given event up to its detection are shown at the top of the figure. (**i**) shows the simplest design, with only one node at each stage. (**ii**) shows a more complex network, featuring two sensors sharing the same sensing environment, each with their own conditioning step. (**iii**) displays the network used in this study, with two sensing environments (EMG and IMU sensing), each with 12 sensors, conditioned individually and combined together in the processing step.

**Figure 3 sensors-22-02603-f003:**
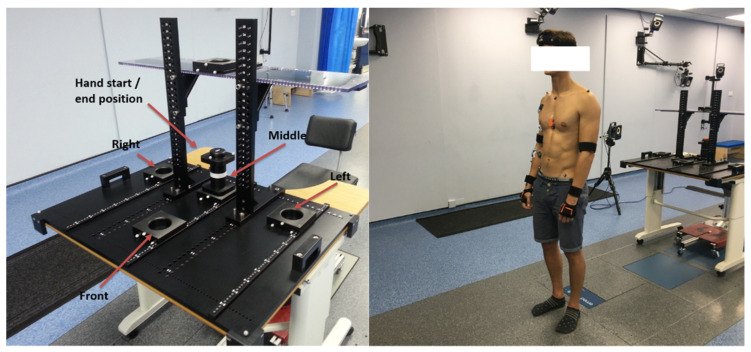
Photographs of the experimental setup used for the acquisition of data used in this study. The configuration pictured is for the reach-grasp activity.

**Figure 4 sensors-22-02603-f004:**
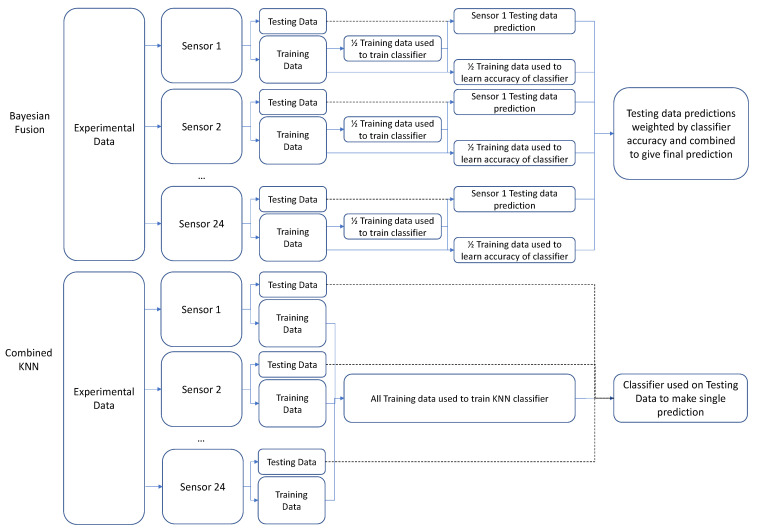
Overall pipeline of training and testing for the two algorithms. The Bayesian Fusion (MM) algorithm divides the training dataset into Classifier Training and Probability Learning subsets. The Classifier Training Set is used to train the classifier for each sensor, and then the Probability Learning set is used to populate the confusion matrix for each classifier. The confusion matrix then provides weightings for each sensor’s contribution to the overall network output, which is used to predict the class of the testing set. The Combined KNN (NMM) does not subdivide the training set, instead using all sensor data to train a single classifier, which then predicts the class of the testing set. Dashed lines indicate testing data.

**Figure 5 sensors-22-02603-f005:**
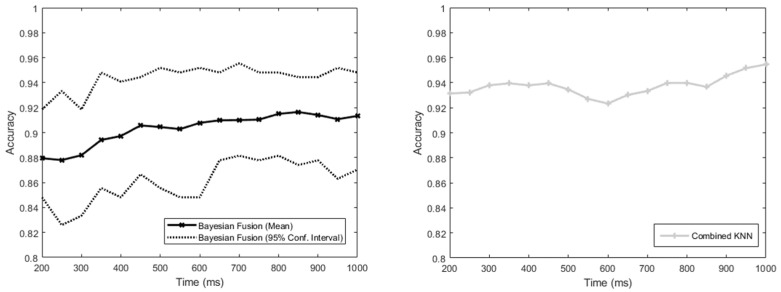
Graphs to show the accuracy of the intent-classification system against the time allowed to pass after the activity’s inception before classification was performed, up to the 1-second limit and with all 24 sensors active (no dropout). These graphs are shown separately to clearly illustrate the presence of a general trend for each algorithm, but comparisons between the two in this context should be avoided (see Section 4.1 for discussion on this). The Combined KNN method does not have confidence intervals, as all the sensors are included and there is no subdivision of the training data, so its performance is entirely reproducible.

**Figure 6 sensors-22-02603-f006:**
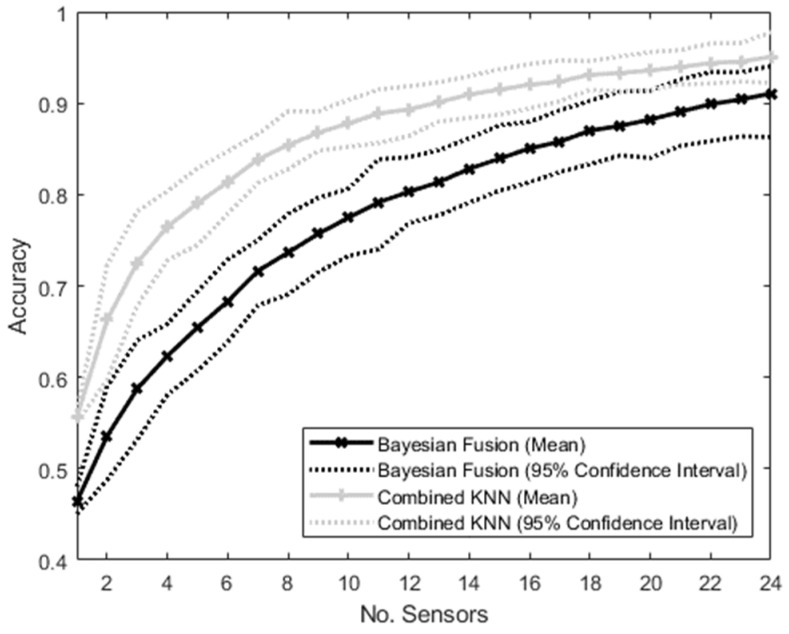
Graph showing the accuracy of the intent-classification system using the Bayesian Fusion method (treating each sensor separately and then combining them) and the combined method (putting all sensor information into a single KNN classifier) as the number of sensors increases. No sensors dropped out—instead, the number of sensors was varied from 1 to 24, and the algorithms were trained on the number of sensors active in each case.

**Figure 7 sensors-22-02603-f007:**
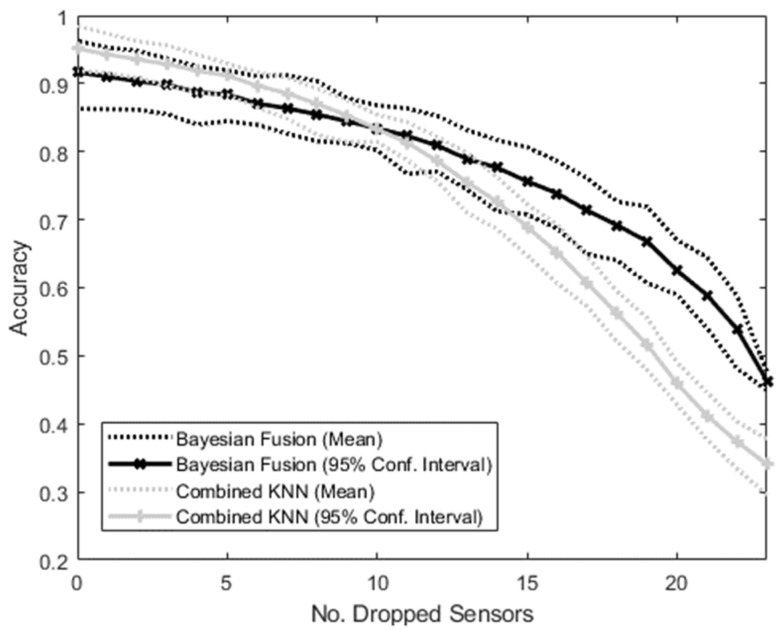
Graph showing the accuracy of the intent-classification system with increasing number of sensors dropping out. The Bayesian Fusion method (treating each sensor separately and then combining them) and the combined method (putting all sensor information into a single KNN classifier) are shown.

**Table 1 sensors-22-02603-t001:** Breakdown of the number of features used from each sensing input available.

Measured Value	No. Features
Orientation	4
Accelerometer	3
Magnetometer	3
sEMG	11
**Total**	**21**

## Data Availability

Data can be obtained on request by contacting the named researchers.

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
