# Peer review of "Towards Dynamic Multi-Modal Intent Sensing Using Probabilistic Sensor Networks"

_sensors, 2022, doi:10.3390/s22072603_

Round 1

Reviewer 1 Report

In this manuscript, the authors proposed a method that treats each sensor individually and combines them using Bayesian sensor fusion. And the objective of this study is to develop and test a dynamic intent sensing system under changing conditions. The work is well written and structured, sufficiently innovative. This is very useful for improving people's quality of life,especially for patients. I suggest minor revision before publication in Sensors as follows:

  1. Line 25-26, use the same symbols between keywords, but here both semicolons and commas appear.
  2. Line 29, “Intent Sensing” is the only one with “1.” in front of it . While all the other secondary headlines,  such as “Sensor Networks for Intent”,  “Probabilistic Sensor Networks”,  “Objective” , there is no such serial number. 
  3. The quality of illustration needs to be improved. For example, the amount of information contained in Figure 1 and Figure 2 is too little, and the two images in Figure 5 are more intuitive when combined into one image.
  4. Page 8, it would make sense to add the relevant formulas of Spearman's rank correlation coefficient to the “Time Variation”.

Author Response

We are grateful to the reviewer for this feedback. The paper has undergone a further editing pass to improve the quality of language and the reviewer's points are addressed as follows:

  1. Semicolons are now consistently used to separate keywords.
  2. Chapter subheadings now all use a numbering system and the text has been updated to reflect this.
  3. Figures 1 and 2 have been updated and now include more detailed captions further elaborating on the information presented. The graphs in Figure 5 have been presented separately to more clearly show the trend in each algorithm - overlaying them appeared to make this more difficult to interpret and encouraged misleading comparisons between them. The aim of Figure 5 is not to compare the algorithms' performance, but to illustrate that both follow a positively increasing trend of accuracy vs. time. A note has been added to the caption explaining this and pointing toward further discussion in section 4.1.
  4. The relevant formula used to calculate the Spearman's rank correlation coefficient has been added on page 9.

We hope that these changes have improved the quality of the paper and look forward to any further correspondence.

Reviewer 2 Report

The authors describe the methodology and background very well. It concerns the presentation of the experimental stand as well. The results are described also well.
In my opinion, the Authors should improve the language. It seems surprising since all Authors are from the University of Oxford. The Abstract and Introduction are very well written, but the language becomes rougher when going further. 

Author Response

We are grateful to the reviewer for this feedback. To address the concerns regarding the quality of writing, the paper has undergone a further editing pass, incorporating language suggestions from two independent readers. It is hoped that the new version is of a more consistent quality suitable for publication.